# Physioxia Has a Beneficial Effect on Cartilage Matrix Production in Interleukin-1 Beta-Inhibited Mesenchymal Stem Cell Chondrogenesis

**DOI:** 10.3390/cells8080936

**Published:** 2019-08-20

**Authors:** Girish Pattappa, Ruth Schewior, Isabelle Hofmeister, Jennifer Seja, Johannes Zellner, Brian Johnstone, Denitsa Docheva, Peter Angele

**Affiliations:** 1Laboratory of Experimental Trauma Surgery, Department of Trauma Surgery, University Hospital Regensburg, Franz Josef Strauss Allee 11, 93053 Regensburg, Germany; 2Department of Orthopaedics and Rehabilitation, Oregon Health & Science University, 3181 SW Sam Jackson Park Rd, OP31, Portland, OR 97239, USA; 3Sporthopaedicum Regensburg, Hildegard von Bingen Strasse 1, 93053 Regensburg, Germany

**Keywords:** cartilage, mesenchymal stem cells, chondrogenesis, hypoxia, interleukin-1β, early osteoarthritis

## Abstract

Osteoarthritis (OA) is a degenerative condition that involves the production of inflammatory cytokines (e.g., interleukin-1β (IL-1β), tumour necrosis factor-alpha (TNF-α) and interleukin-6 (IL-6)) that stimulate degradative enzymes, matrix metalloproteinases (MMPs) and aggrecanases (ADAMTS) resulting in articular cartilage breakdown. The presence of interleukin-1β (IL-1β) is one reason for poor clinical outcomes in current cell-based tissue engineering strategies for treating focal early osteoarthritic defects. Mesenchymal stem cells (MSCs) are a potential cell source for articular cartilage regeneration, although IL-1β has been shown to inhibit in vitro chondrogenesis. In vivo, articular chondrocytes reside under a low oxygen environment between 2–5% oxygen (physioxia) and have been shown to enhance in vitro MSC chondrogenic matrix content with reduced hypertrophic marker expression under these conditions. The present investigation sought to understand the effect of physioxia on IL-1β inhibited MSC chondrogenesis. MSCs expanded under physioxic (2% oxygen) and hyperoxic (20%) conditions, then chondrogenically differentiated as pellets in the presence of TGF-β1 and either 0.1 or 0.5 ng/mL IL-1β. Results showed that there were donor variations in response to physioxic culture based on intrinsic GAG content under hyperoxia. In physioxia responsive donors, MSC chondrogenesis significantly increased GAG and collagen II content, whilst hypertrophic markers were reduced compared with hyperoxia. In the presence of IL-1β, these donors showed a significant increase in cartilage matrix gene expression and GAG content relative to hyperoxic conditions. In contrast, a set of MSC donors were unresponsive to physioxia and showed no significant increase in matrix production independent of IL-1β presence. Thus, physioxia has a beneficial effect on MSC cartilage matrix production in responsive donors with or without IL-1β application. The mechanisms controlling the MSC chondrogenic response in both physioxia responsive and unresponsive donors are to be elucidated in future investigations.

## 1. Introduction

Osteoarthritis (OA) is a degenerative condition involving changes in articular cartilage matrix resulting from the stimulation of matrix metalloproteinases (MMPs) and aggrecanases (a disintegrin and metalloproteinase with thrombospondin motifs, ADAMTS) [1,2]. Focal degenerative defects in the early stages of OA can be treated via autologous chondrocyte implantation (ACI), although studies have shown that there is a reduced improvement in clinical outcome scores and higher probability for graft re-operation (two-fold failure rate) compared to traumatic defects [3,4]. A reason for the poor outcomes in degenerative defects is the presence of inflammatory cytokines including interleukin-1alpha (IL-1α), interleukin-1beta (IL-1β), and tumour necrosis factor-alpha (TNF-α) [5,6,7,8,9,10,11,12]. Specifically, in an early OA situation, the presence of IL-1β has been shown to negatively influence the clinical outcome of ACI for the treatment of degenerative cartilage lesions [3,4]. 

Alternative cell sources and culture conditions that can enable matrix production in spite of cytokine presence are required to ensure successful clinical outcomes, especially for early OA lesions. Mesenchymal stem cells (MSCs) are a suitable cell source for therapies due to their multilineage differentiation potential, particularly towards the chondrogenic lineage [13,14]. However, IL-1β has been shown to have an inhibitory effect on MSC chondrogenesis [15,16]. In vitro studies examining methods to reduce the detrimental response have used anti-arthritic neutraceuticals (e.g., curcumin, reservatrol, mangiferin), adenoviruses (e.g., lL-1 receptor antagonist) or mechanical stimulation (e.g., ultrasound) [16,17,18,19,20,21,22,23]. The latter stimuli show that the negative effects of IL-1β on MSC chondrogenesis can be reduced via environmental stimuli.

In vivo, articular chondrocytes reside under an oxygen tension between 2–5% oxygen that is referred to as physioxia (the physiological oxygen tension) for these cells. Thus, the atmospheric oxygen conditions (20% oxygen) of standard tissue culture incubators provide a hyperoxic state for chondrocytes [24,25]. In vitro culture of articular chondrocytes under physioxia has been shown to enhance cartilage matrix synthesis compared with hyperoxic culture [26,27,28,29,30,31,32,33,34,35,36]. Similar results were found for MSCs [37,38,39,40,41,42,43,44]. However, whilst early data provided contradictory evidence as to whether physioxia affects hypertrophy in MSCs, more recent data indicate a downregulation in the expression of hypertrophic markers (e.g., MMP13 and collagen X) during physioxic MSC chondrogenesis [37,39,40,41]. The present investigation sought to understand the effect of physioxia on MSC chondrogenesis under normal conditions and in the presence of IL-1β. Based on the previous literature, it was hypothesized that physioxia would provide a beneficial response that would counter the effects of IL-1β-inhibited MSC chondrogenesis. 

## 2. Materials and Methods

### 2.1. Human MSC Isolation and Harvesting

Human bone marrow aspirates were isolated from the iliac crests of nine male patients (mean age: 30 ± 11 years) undergoing knee arthroplasty and that required autologous bone grafting, sourced from the iliac crest for the treatment of deep osteochondral defects, following informed consent of the patients and using procedures that were approved by the local ethics committee (University Hospital Regensburg; Ethic approval no.: Nr. 00/134) [45]. The mononuclear cell population from bone marrow aspirates was counted and then seeded into flasks at a density of 1 × 10^5^ cells/cm^2^ in low glucose Dulbecco’s Modified Eagle medium (DMEM; Invitrogen, Karlsruhe, Germany) supplemented with 10% (*v*/*v*) foetal bovine serum (FBS; PAN Biotech, Aidenbach, Germany), 1% (*v*/*v*) penicillin-streptomycin (Invitrogen) and 5 ng/mL basic fibroblastic growth factor (bFGF; Peprotech, Hamburg, Germany). Flasks were cultured in parallel in either a standard cell culture incubator at 20% oxygen, 5% CO_2_ and 70% N_2_ or a low oxygen incubator (ThermoFisher Scientific, Regensburg Germany) set at 2% oxygen, 5% CO_2_ and 93% N_2_. For this manuscript, hyperoxia refers to 20% oxygen, whilst 2% oxygen is termed as physioxia [24,39] The first media change was performed after 5 days and then media was replenished twice a week until trypsinisation at 80% confluence and further passaged. Previous publications using this method of isolation have demonstrated that the MSC population has a negative expression of CD34, CD45 and CD19, whilst they showed positive expression of CD44, CD73 CD90, CD105 and CD166 [37,46,47,48]. Cells were used for experiments at no later than passage 3.

### 2.2. Chondrogenic Differentiation

Monolayer-expanded MSCs cultured under either hyperoxia or physioxia were used to form pellet cultures as previously described [14]. In brief, pellets were formed by centrifuging 2 × 10^5^ MSCs at 250× *g* for 5 min in 300 µL chondrogenic medium in polypropylene V-bottom 96-well plates. Chondrogenic media consisted of serum-free high-glucose DMEM containing 10 ng/mL TGF-β1 (R&D systems), 100 nM dexamethasone, 50 µg/mL ascorbic acid-2-phosphate (all Sigma-Aldrich, Steinheim, Germany), 1 mM sodium pyruvate (Invitrogen) and 1% ITS (PAN Biotech GmbH, Aidenbach, Germany). An initial experiment was performed to observe the effect IL-1β on MSC chondrogenesis. Hyperoxic MSC chondrogenic pellets were cultured in the presence of 0.1, 0.5, 1 and 10 ng/mL interleukin-1beta (IL-1β; Peprotech). The range of concentrations chosen for the study were based on previous literature regarding the levels of IL-1β during osteoarthritis [8]. Following this study, chondrogenic media supplemented with either 0.1 or 0.5 ng/mL IL-1β were applied to both hyperoxic and physioxic MSC pellets. Pellets were then cultured under their respective expansion oxygen conditions for 21 days with media changes performed every 2–3 days. In the case of physioxia pellets, media was pre-equilibrated in a physioxia incubator prior to replenishment.

### 2.3. Wet Weight and GAG Assay

After 21 days in culture, the wet weight of pellets was measured. Media was collected at each feeding during the culture period and included in the GAG measurements. Triplicate pellets from each group were then digested with 150 µg/mL papain in PBS, pH 6.0, containing 8 mM sodium EDTA, 6 mM L-cysteine (all Sigma-Aldrich). Sulfated glycosaminoglycan (GAG) and DNA content were quantified using 1,9-dimethylene blue (DMMB) and Picogreen (Quant-iT dsDNA; Invitrogen, Carlsbad, CA, USA) assays, respectively. Digested pellet GAG and collected supernatant were quantified against a standard curve generated using bovine chondroitin sulphate A (Sigma-Aldrich) diluted in either DMEM or papain buffer as standard in serial dilution. DMMB dye (18 µg/mL in 0.5% ethanol, 0.2% formic acid, 30 mM sodium formate, pH 3) was added to standards and samples and absorbance measured at 575 nm (Tecan, Crailsheim, Germany). DNA content in pellet digests was quantified using Quant-iT dsDNA assay according to manufacturers’ instructions. 

### 2.4. Collagen I and II ELISA

Six pellets per condition were taken from culture on day 21 and homogenized using a PreCellys homogenizer (Bertin Instruments, Montigny le Bretonneux, France), then digested using 10 mg/mL pepsin in 0.05 M acetic acid, 0.5 M NaCl, pH 2.9 with continuous shaking at 4 °C for 48 h. Subsequent steps and ELISA were performed according to manufacturer’s protocol (Type I Collagen detection kit, Type II Collagen detection kit; both from Chondrex, Redmond, WA, USA).

### 2.5. Histology and Immunohistochemistry

Pellets were fixed in 4% PBS buffered paraformaldehyde, rinsed briefly in PBS and then incubated in increasing sucrose concentrations (10–30%). Pellets were photographed with an optical microscope (PL2000, Optech, Germany) and then embedded in Tissue-Tek (Sakura, Zoeterwnde, The Netherlands). Embedded pellets were cryosectioned at 10 µm with a HM500 OM cryotome (Microm, Berlin, Germany). Sulphated glycosaminoglycan content was observed by histochemical staining with DMMB (0.05% 1,9-dimethylmethylene blue, 0.5% ethanol, 0.2% formic acid, 30 mM sodium formate, pH 3).

Sections used for immunohistochemistry were rehydrated and antigen retrieval was performed at room temperature. For collagen II (Calbiochem, Darmstadt, Germany) and MMP-13 (Abcam, Cambridge, UK), sections were treated with 3 mg/mL pepsin (Sigma-Aldrich) in 1× citric/phosphate McIlvaine buffer for 15 min. Sections used for collagen X (X53, ThermoFisher Scientific) staining were treated with 1 mg/mL hyaluronidase in PBS (pH 5) for 60 min at 37 °C followed by pepsin treatment as previously described. All sections were blocked with a blocking buffer (10% goat serum in 1× PBS (supplemented with 10% FBS for collagen II antibody)) for 1 h and then probed for human collagen II (mouse monoclonal antibody, 1:200; Calbiochem), MMP-13 (rabbit polyclonal antibody, 1:200) and collagen X (mouse monoclonal antibody, 1:50) in blocking buffer during overnight incubation with gentle rocking. Following this, sections were washed and then secondary antibodies, fluorescein (FITC) conjugated anti-rabbit IgG (1:200) and fluorescein (FITC) conjugated anti-mouse IgG (1:200) (Jackson Immuno Research, Cambridge, UK) diluted in blocking buffer were applied and incubated for 1 h. Slides were incubated with 4,6-diamidino-2-phenylindole (DAPI; 1:100,000 in PBS) for 5 min and then mounted with Mowiol anti-fading media and imaged using an Olympus XC10 camera on an Olympus BX61 fluorescence microscope (Olympus, Japan). 

### 2.6. Gene Expression Analysis

RNA was extracted from a pool of six pellets for each condition or treatment. Pellets were snap frozen, QIAzol (Qiagen, Hilden, Germany) added and then homogenized using a PreCellys system homogenizer. Solution was removed and RNA was isolated using QIAzol method according to the manusfacturer’s instructions. Samples were purified and genomic DNA removed via RNeasy Mini kit (QIAGEN) according to manufacturers’ instructions. Total RNA was quantified and 200 ng was reverse-transcribed using Transcriptor first strand kit (Roche, Mannheim, Germany). Quantitative polymerase chain reaction (qPCR) of chondrogenic genes (listed in Table 1) was performed using Real Time PCR Ready Custom designed plates (BioRad Laboratories, Munich, Germany) according to the manufacturer’s instructions [49]. qPCR reactions were performed using a Biorad CFX96 system (Biorad). Results were analysed using the ΔΔCt method and normalized to the ribosomal protein 13a (RPL13a)—the most stably expressed of the three housekeeping genes (hypoxanthine phosphoribosyltransferase (HPRT), Proteasome subunit beta type-4 (PSMB4) and RPL13a) evaluated [50]. Gene expression in physioxia was calculated as fold change from that measured under hyperoxia. 

### 2.7. Statistical Analysis

All statistical analysis was performed using Graphpad Prism v7 (GraphPad, La Jolla, CA, USA). A comparison of pellet wet weight, GAG and collagen II content between hyperoxia and physioxia was performed using two-way ANOVA with Tukey post-hoc test, with significance set at *p* < 0.05. Gene expression data was calculated as fold change in physioxia relative to hyperoxia either with or without IL-1β and then analysed using a Mann-Whitney test with significance set at *p* < 0.05. 

## 3. Results

### 3.1. IL-1β Shows a Dose Dependant Decrease in Chondrogenic MSC Pellet Wet Weight and GAG Content

To understand the effect and choose appropriate conditions for examination under physioxia, IL-1β concentrations ranging from 0.1–10 ng/mL were applied to chondrogenic MSCs under hyperoxia. Results showed a significant decrease in both pellet wet weight and GAG content in the presence of IL-1β relative to control chondrogenesis at concentrations above 0.1 ng/mL (Figure 1a,b; * *p* < 0.05). There were no significant differences in wet weight or GAG content between pellets subjected to concentrations above 0.5 ng/mL IL-1β. Macroscopic and DMMB histological images correlated with these results, whereby at 0.1 ng/mL IL-1β, pellets were smaller but stained for glycosaminoglycans within the matrix. At concentrations at or greater than 0.5 ng/mL IL-1β, there was no GAG deposition in the pellet and all were of similar size (Figure 1c). Thus, 0.1 and 0.5 ng/mL IL-1β concentrations were used for physioxia experiments. 

### 3.2. Physioxia Alone Enhances MSC Chondrogenic Matrix Expression and Content

MSC chondrogenesis without the presence of IL-1β was first performed to understand the basic response under physioxia. Examination of all donors indicated a significant difference in matrix production under physioxia, with respect to wet weight (*p* = 0.0004), GAG (*p* < 0.0001) and collagen II (*p* = 0.0078) content (Figure 2a–c; * *p* < 0.05). The analysis demonstrated that some donors within the cohort examined, did not respond to physioxia. A previous publication demonstrated that MSC donors can be separated into two discrete groups, based on their GAG production under hyperoxia relative to chondrocytes under the same conditions (Figure 2d) [39]. Using the data from the publication, thresholds for high or low GAG donors (indicated by the dotted lines) were calculated and then used to disseminate the donors into physioxia responsive and non-responsive donors (Figure 2d). Donors that were either below or above the first threshold were known as low GAG donors or physioxia responders, whereas those at or above the second threshold are described as high GAG or physioxia non-responders. The latter may also be described as a donor with less than a two-fold increase in GAG content under physioxia. 

In the cohort for this study, there were two non-responding donors (20% of all donors examined) that showed no difference in pellet wet weight, GAG and collagen II content compared with hyperoxic cultures (Figure 3a,c,e). This was supported by histological analysis of these pellets whereby, there was no increase in pellet size, or enhancement in DMMB and collagen II staining under physioxia (Figure 3b,d,f). In contrast, responders showed a significant increase in pellet wet weight and GAG (eight-fold increase; both *p* < 0.0001) and collagen II content (three-fold increase; *p* = 0.0121) with respect to hyperoxia (Figure 4a,c,e; * *p* < 0.05). In contrast, analysis of collagen I in these responsive donors demonstrated that physioxic MSC chondrogenesis had a significantly reduced collagen I content (four-fold decrease; *p* = 0.0022) compared to hyperoxic MSCs (Figure 4g: * *p* < 0.05). Furthermore, the ratio between collagen II to collagen I was greater under physioxia (mean ratio: 15 ± 10) compared to hyperoxic (mean ration: 1 ± 0.5) MSC chondrogenesis. Macroscopic, DMMB and collagen II histology correlated with these results, whereby larger pellets, greater metachromasia and collagen II staining was observed under these conditions (Figure 4b,d,f,h). 

### 3.3. Pellet Wet Weight and GAG Content in IL-1β Inhibited Chondrogenesis Is Suppressed under Physioxia in Responsive Donors

In non-responsive donors, physioxia did not significantly increase the wet weight, GAG and collagen II content in IL-1β treated pellets with histological staining supporting these results (Figure 3). In contrast, physioxia-responding donors had significantly higher wet weight (*p* = 0.0433) and GAG content (*p* = 0.0267) in the presence of 0.1 ng/mL IL-1β relative to corresponding pellets in hyperoxia (Figure 4a,c; * *p* < 0.05). Histological images supported these results, whereby pellet size and DMMB staining were greater in physioxic conditions (Figure 3b,d). Interestingly, there was no significant increase in collagen II content in IL-1β-treated pellets under physioxia, with histological images in agreement with these findings (Figure 4e,f). Furthermore, there was a significant increase in collagen I content compared to the respective control conditions, particularly under physioxia (0 vs. 0.1 ng/mL IL-1β, *p* = 0.0118) and between 0 ng/mL and 0.1 ng/mL IL-1β under hyperoxia (*p* = 0.0092; Figure 4g,h). Physioxia showed a tendency to reduce collagen I content in IL-1β-treated pellets compared to hyperoxic pellets but there was no significant difference either on a protein level or staining between corresponding IL-1β-treated conditions (Figure 4g,h).

### 3.4. Physioxia Enhances Gene Expression of Chondrogenesis-Associated Markers and Suppresses Markers for Late Stage Hypertrophy

The gene expression for control and IL-1β-treated pellets in responsive donors was examined to determine how cartilage transcription factors, TGF-β receptors, matrix (Figure 5) and hypertrophy genes were regulated under physioxia (Figure 6). For these studies, only 0.1 ng/mL IL-1β pellets were analysed due to the similar effects on physioxia on IL-1β chondrogenesis. Physioxic culture promoted SOX gene expression (Figure 5a). Conducting chondrogenesis under physioxia also lowered the negative effects of IL-1β, whereby cartilage transcription factors, SOX5, 6 and 9 and TGF-β receptor I and II (for all genes stated, *p* = 0.0286) were all significantly upregulated for physioxic chondrogenesis compared with hyperoxic cultured pellets (Figure 5b, * *p* < 0.05). Cartilage matrix protein genes examined were also significantly upregulated under physioxia, with or without IL-1β (Figure 5c,d; * *p* < 0.05) including aggrecan and collagen II that correlate with the protein data (Figure 2, Figure 3 and Figure 4). 

Analysis of hypertrophic markers in physioxia chondrogenesis showed that in both non-responsive and responsive donors, there was a suppression in collagen X (Figure 6a,c) and MMP13 protein expression (Figure 6b,d) under physioxic conditions, particularly under control and 0.1 ng/mL IL-1β, whereas there was greater expression at the corresponding conditions under hyperoxia. However, contrary to these results, the analysis of these early hypertrophic genes in responsive donors showed an upregulation in collagen X and MMP-13 expression under physioxia, independent of the presence of IL-1β, with collagen X (*p* = 0.0286) being significantly upregulated (Figure 6e,f; * *p* < 0.05). However, physioxic chondrogenesis alone showed a significant downregulation in late-stage hypertophic markers, specifically the aggrecanases (ADAMTS-4 and ADAMTS-5) and the osteogenic transcription factor, RUNX2 (for all genes stated, *p* = 0.0286) (Figure 6e; * *p* < 0.05), whereas in the presence of IL-1β, there was only minimal expression of these genes (Figure 6f: * *p* < 0.05). 

## 4. Discussion

The aim of the present investigation was to understand how physioxia modulates MSC chondrogenesis and specifically, its effect on IL-1β-inhibited chondrogenesis. Under control conditions, there was found to be donors that were responsive and non-responsive to physioxic conditions (Figure 2). In this instance, two non-responsive donors were found amongst those examined and showed no difference in pellet wet weight, GAG and collagen II content under physioxia with or without IL-1β presence (Figure 3). It was found that under both control and in the presence of IL-1β, responsive donors demonstrated a significant increase in wet weight and GAG content that corresponded to an upregulation in cartilage transcription factors and matrix genes under physioxic conditions (Figure 4 and Figure 5). 

It has been well-established that physioxic conditions enhance the anabolic response in MSC chondrogenesis [15,37,39,40,41,42,43,51,52]. Anderson et al. (2016) showed that the reaction under physioxia is related to the GAG production under hyperoxia. Using the data from this publication, thresholds for low and high GAGs were calculated and then used to separate donors accordingly. High GAG donors or physioxia non-responders did not show an increase in GAG and collagen II content under physioxia (Figure 3), whereas low GAG or physioxia responsive donors demonstrated significant increases in GAG and collagen II deposition under physioxia with concomitant upregulation in cartilage transcription factors and matrix genes, supporting the data from earlier studies (Figure 4 and Figure 5) [15,37,39,40,41,42,43,51,52]. 

In both donor response types, collagen X and MMP13 protein levels were reduced under physioxia compared to hyperoxia with or without IL-1β presence on day 21, despite an upregulation in gene expression (Figure 6). However, on day 14, COL10A1 gene expression under physioxia was lower compared to day 21 with a corresponding reduction in collagen X protein levels (Appendix A). In comparison, day 14 hyperoxic pellets stained for collagen X in the matrix both with and without the presence of IL-1β. These results indicate that physioxia delays the onset of hypertrophy in MSC chondrogenesis, as protein expression occurs at a later time point. This has been demonstrated in a previous publication, whereby protein expression for hypertrophic markers was shown on day 28 physioxic chondrogenic pellets and reduced compared to those under hyperoxia [53]. 

In the present investigation, physioxia responsive donors still had a significant increase in wet weight and GAG content over pellets in hyperoxia when cultured in the presence of IL-1β, and gene expression results were consistent with these data (Figure 4 and Figure 5b,d). These data are supported by a previous publication by Felka et al. (2009) under similar experimental conditions, whereby physioxia helped to reduce the negative effects of IL-1β on MSC chondrogenesis, although this was a more limited study that only performed gene expression and histological analysis [15]. Boeuf et al. (2012) showed that the effects of IL-1β on MSC chondrogenesis included the stimulation of aggrecanolysis [54]. Our study demonstrated that physioxia downregulated the aggrecanases, ADAMTS-4 and ADAMTS-5. However, a concomitant rise in collagen II production in physioxia with IL-1β treatment was not seen. TNF-α, another pro-inflammatory cytokine involved in OA, has been shown to induce collagen II degradation via MMPs and that physioxia could reduce this effect. Furthermore, the described TNF-α model also showed an increased GAG content and reduced ADAMTS expression under physioxia, similar to the data shown in this study using IL-1β [34]. Thus, different cytokines initiate distinct degenerative cascades that can be influenced by pro-anabolic stimuli. Models utilising multiple cytokines (e.g., TNF-α, IL-6 and IL-13) would provide a more accurate analysis of OA progression and methods to reduce its degenerative effect on MSC chondrogenesis [5,6,7,8,9,10,11,12]. 

An interesting finding within this study is that even in the presence of IL-1β, physioxia non-responsive or high GAG donors did not show an increase in wet weight, GAG and collagen II content under physioxia (Figure 2). In spite of the low donor number, the presence of donors that are non-responsive to physioxia indicates that MSC chondrogenesis is a donor-dependant process. Markers predicting whether a donor has an anabolic response under physioxia remain to be elucidated. Furthermore, these non-responsive physioxia donors did not show an increase in GAG content or wet weight under physioxia in the presence of IL-1β, suggesting that there are discrete mechanisms occurring between physioxia responsive and non-responsive donors both in the absence and presence of IL-1β. A greater cohort of donors from the described groups is required to understand the mechanisms controlling the response, both with respect to monolayer expansion and pellet culture. 

A potential mechanism for restoring MSC chondrogenesis in the presence of IL-1β-inhibited chondrogenesis is TGF-β receptor I/II, as these were upregulated under physioxia (Figure 5b) [55]. IL-1β inhibition in chondrocytes has been shown to involve the loss of TGF-β receptor I/II activity, resulting in downregulation of SOX9 [56,57]. [16]. Downstream of the receptor, IL-1β inhibits phosphorylation of SMAD3/4 in TGF-β-stimulated chondrocytes, whilst concurrently enhancing the inhibitory SMAD7 pathway [58]. It may be postulated that physioxia restores SMAD3/4 phosphorylation in spite of IL-1β presence and is a downstream pathway that rescues MSC chondrogenesis in the presence of inflammatory cytokines. Studies using chondrocytes have also implicated the NF-κB pathway in cartilage degeneration [16,56,58]. Wehling et al. (2009) demonstrated that silencing this pathway via an adenoviral transduction of the suppressor of NF-κB abrogated TNF-α and IL-1β inhibited MSC chondrogenesis and restored cartilage matrix production [16]. Both the TGF-β/SMAD and NF-kB pathways are to be investigated in future studies. 

The beneficial response induced by physioxia preconditioned MSC chondrogenesis in the presence of IL-1β in responsive donors, means that this is a potential treatment option for focal OA defects. Animal models that mimic post-traumatic or early OA situations are required to understand whether physioxia preconditioned MSCs or chondrogenic implants can regenerate articular cartilage in a clinical setting. In vivo studies using physioxia preconditioned MSC chondrogenic implants (MSCs expanded and in vitro chondrogenically stimulated under physioxia) in animal models have focussed on post-trauma models and shown both an improvement and no difference in cartilage regeneration compared to hyperoxic MSCs [59,60]. In particular, Leitjen et al. (2014) showed that physioxia preconditioned MSC chondrogenic implants produced more stable articular cartilage upon implantation in a nude mouse model, whereas hyperoxic-preconditioned MSC chondrogenic implant became vascularized and formed bone [41]. In contrast, studies using rabbit and sheep models showed no difference in cartilage regeneration [59,60]. However, no studies have evaluated treatment in an early OA situation. In vivo models reflecting this setting are currently in progress. 

There are limitations in the present study that should be noted when evaluating the data. The present study only utilised a hypoxia incubator for maintaining physioxic culture of MSCs and chondrogenic pellets and was not kept continuously under a physioxic environment using a glove-box culture hood. However, this and other investigations that have demonstrated the anabolic effects of physioxia on MSC chondrogenesis, show an enhancement both with [39] and without [37] the use of a hypoxia chamber. Furthermore, there are also no significant differences between 5% and 2% oxygen in previous studies with respect to the beneficial effects of physioxia on chondrogenesis [61]. Thus, our results are in keeping with the literature on physioxia MSC chondrogenesis.

In conclusion, there is a donor-dependant response in MSC chondrogenesis under physioxia, whereby, in responsive donors, there was enhanced cartilage matrix production and a reduced expression of late-stage hypertophic markers. In the presence of IL-1β, physioxia responsive donors showed a significant enhancement in GAG content and an upregulation in cartilage-related genes including cartilage-specific transcription factors and TGF-β receptors that enable MSC chondrogenic restoration. The mechanisms involved in the different responses under physioxia with or without IL-1β and evaluation of the cartilage formation by physioxia-preconditioned MSCs in animal models are to be examined in future studies. 

## Figures and Tables

**Figure 1 cells-08-00936-f001:**
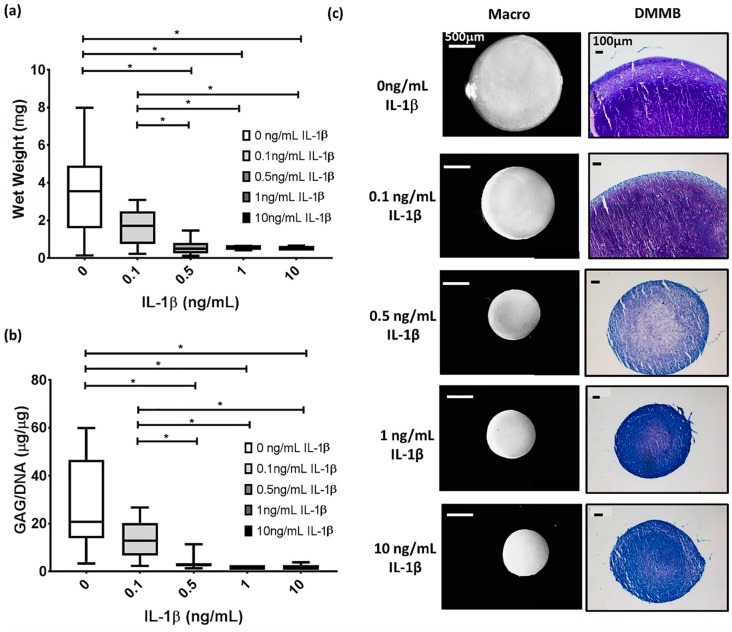
IL-1β dose response under hyperoxic conditions during chondrogenesis, examined for pellet (**a**) wet weight and (**b**) GAG content. Box plots represent median ± I.Q.R; n = 3–12 donors; * *p* < 0.05; (**c**) Representative macroscopic and DMMB-stained chondrogenic pellets under the range of IL-1β concentrations used.

**Figure 2 cells-08-00936-f002:**
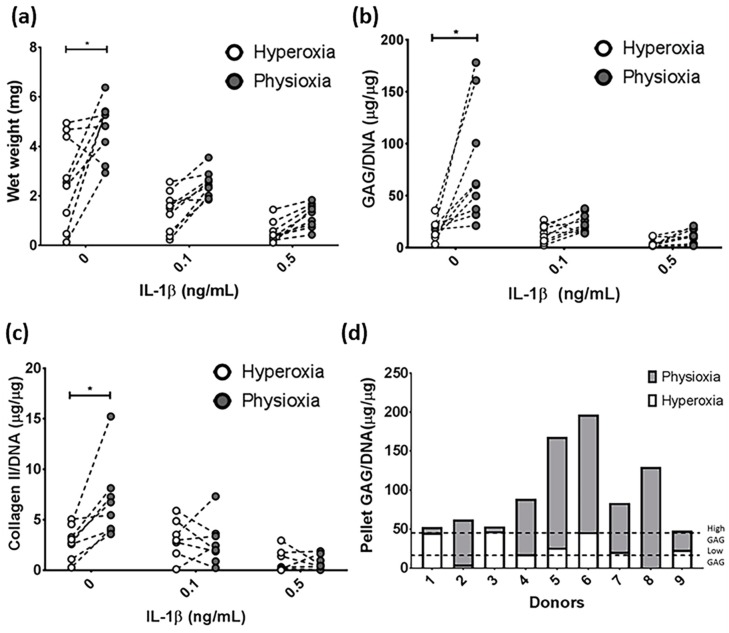
Donor-dependent responses in pellet (**a**) wet weight, (**b**) GAG and (**c**) collagen II content between hyperoxia and physioxia cultured chondrogenic pellets in the presence of IL-1β. Dotted lines represent corresponding donors at hyperoxia and physioxia. Data are presented for nine independent donors (* *p* < 0.05). (**d**) Pellet GAG content for chondrogenic pellets under hyperoxia and physioxia. Dotted lines indicate threshold for low GAG and high GAG donors based on data from a previous publication [39].

**Figure 3 cells-08-00936-f003:**
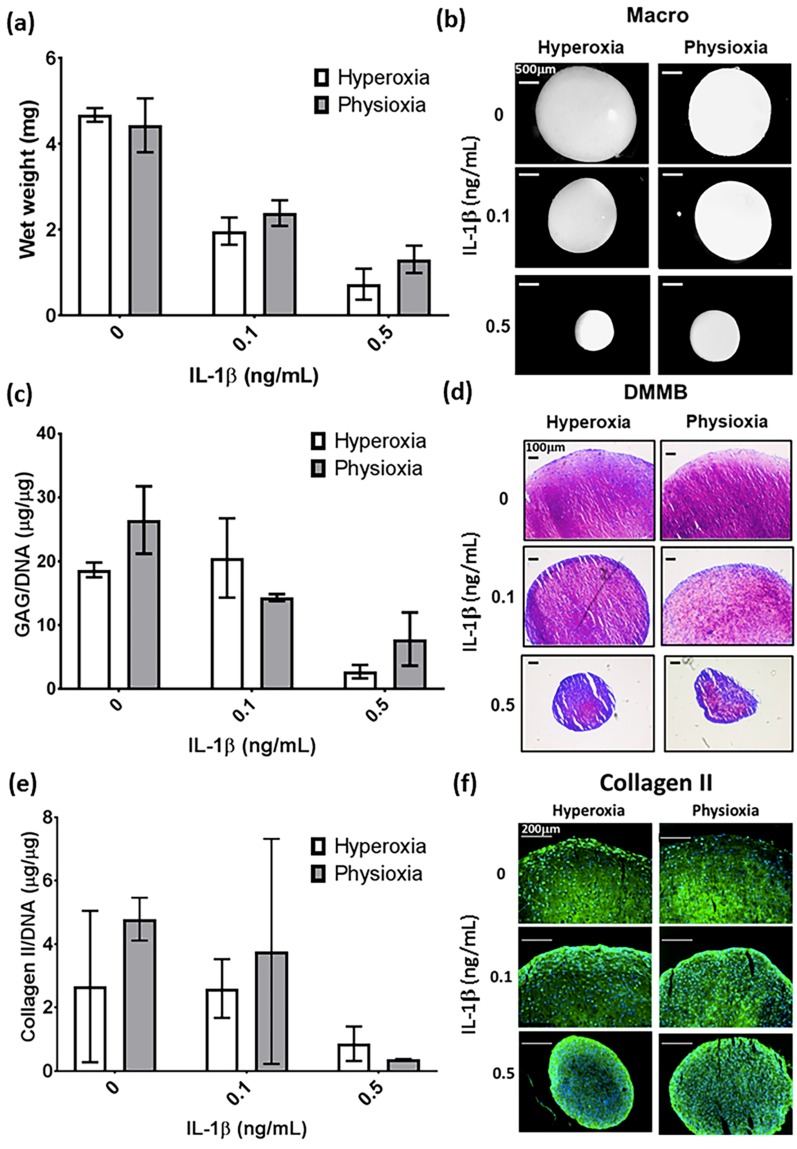
Pellet (**a**) wet weight, (**c**) GAG and (**e**) collagen II content in physioxia non-responsive donors in the presence of IL-1β. Data represent mean ± S.E.M. of n = 2 donors. Representative (**b**) macroscopic, (**d**) DMMB and (**f**) collagen II-stained pellets of a non-responsive donor under physioxia and hyperoxia.

**Figure 4 cells-08-00936-f004:**
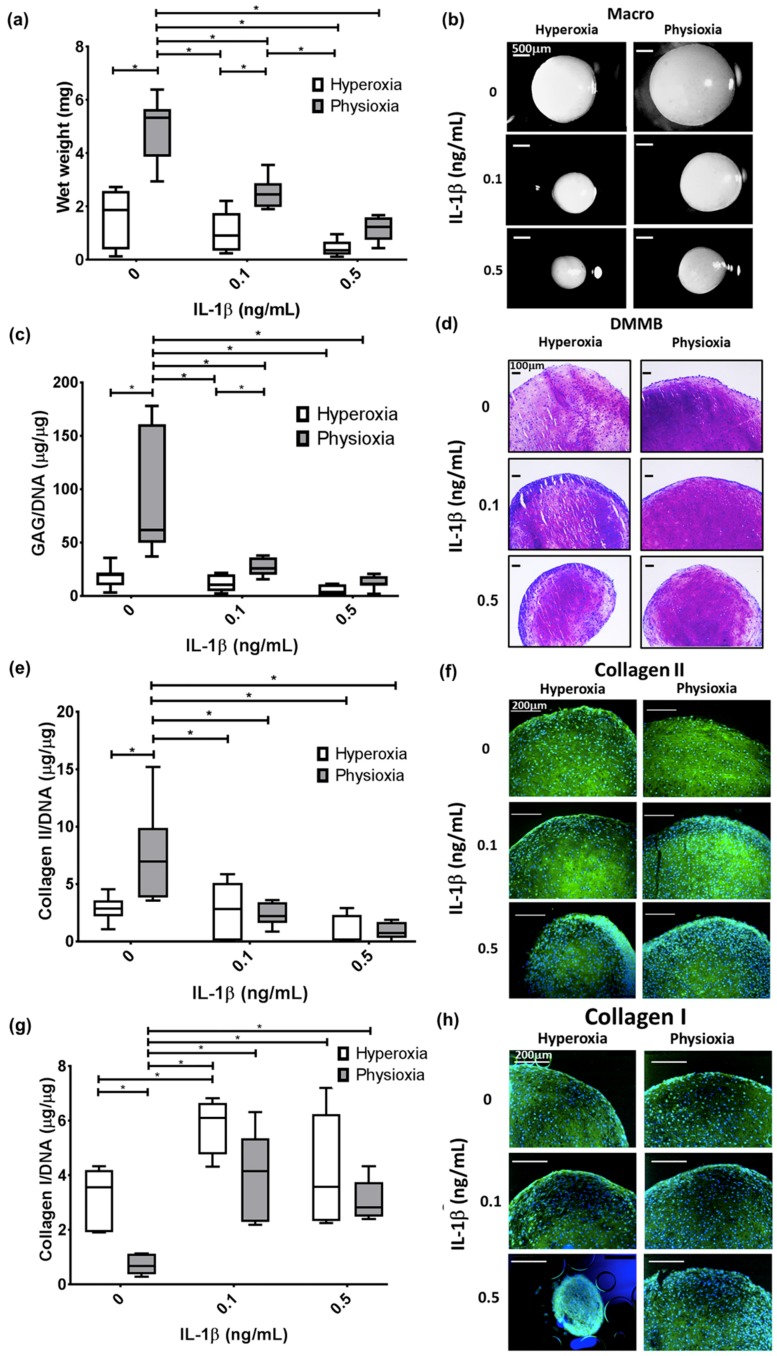
Pellet (**a**) wet weight, (**c**) GAG, (**e**) collagen II and (**g**) collagen I content in physioxia responsive donors in the presence of IL-1β. Box plot represent median ± I.Q:R: of n = 7 donors; * *p* < 0.05. Representative (**b**) macroscopic, (**d**) DMMB, (**f**) collagen II and (**h**) collagen I-stained pellets of a responsive donor under physioxia and hyperoxia.

**Figure 5 cells-08-00936-f005:**
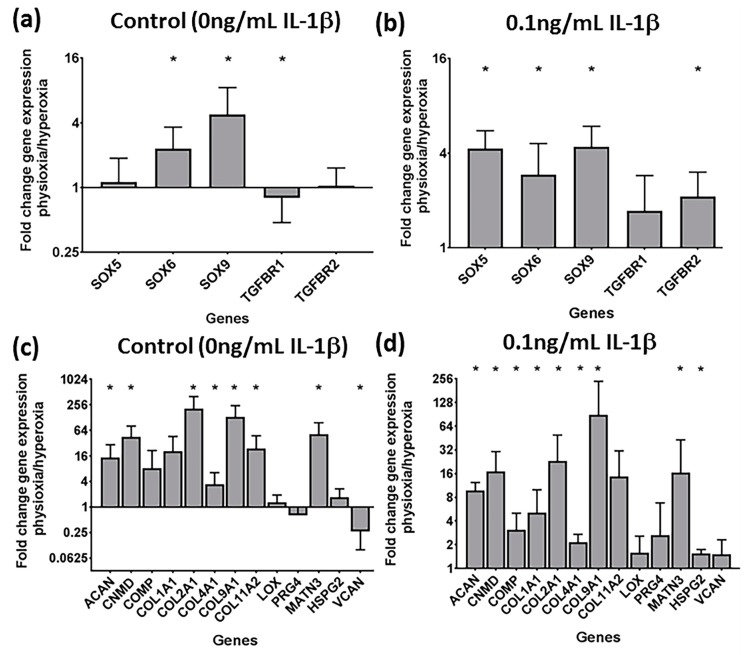
Gene expression of (**a**,**b**) chondrogenic transcription factors and (**c**,**d**) cartilage matrix proteins under physioxia for (**a**,**c**) control and in the presence of (**b**,**d**) 0.1 ng/mL IL-1β for physioxia responsive donors. Data represents the fold change in expression of pellets cultured under (**a**,**c**) 0 ng/mL and (**b**,**d**) 0.1 ng/mL IL-1β physioxia relative to the corresponding conditions under hyperoxia. Data represent mean ± S.D. of n = 4 physioxia responsive donors; * *p* < 0.05.

**Figure 6 cells-08-00936-f006:**
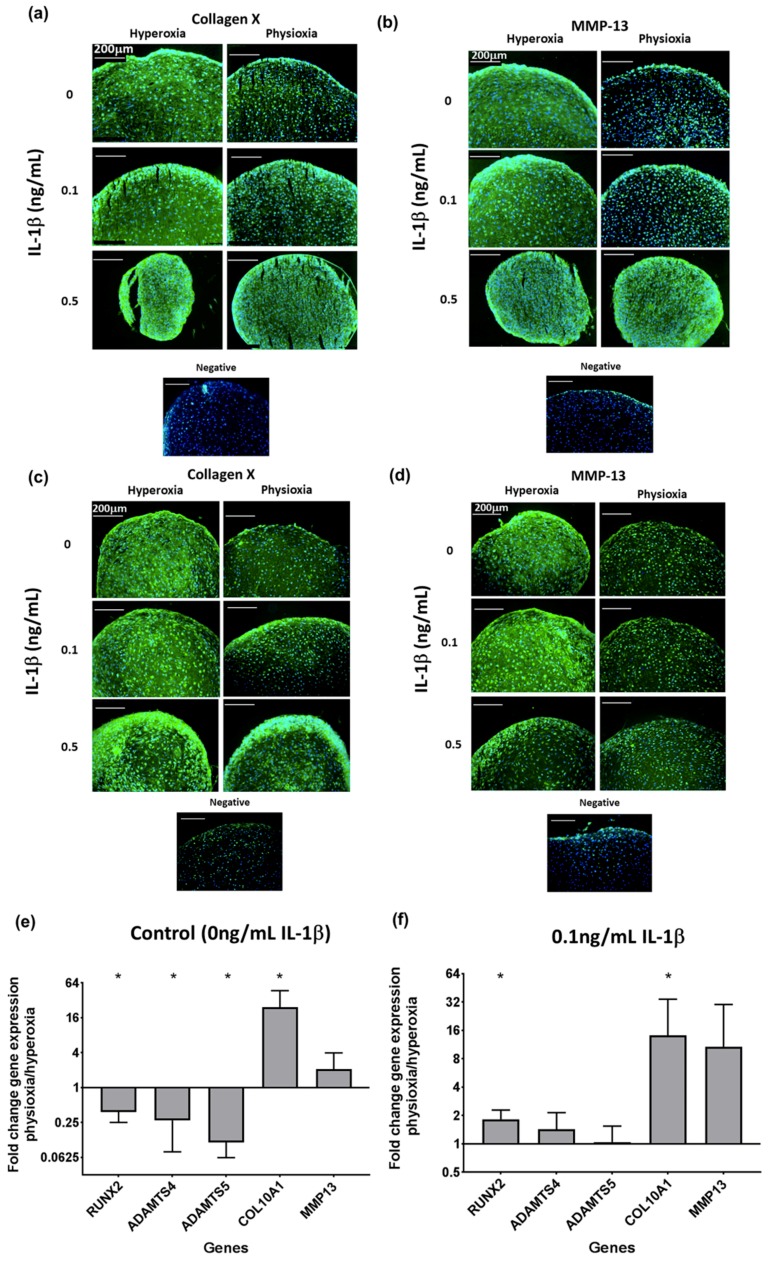
Representative images of chondrogenic pellet stained for (**a**,**c**) collagen X and (**b**,**d**) MMP-13 in physioxia (**a**,**b**) non-responsive and (**c**,**d**) responsive donors at control, 0.1 ng/mL or 0.5 ng/mL IL-1β. Gene expression of (**e**,**f**) hypertophic genes under physioxia for (**e**) control and in the presence of (**f**) 0.1 ng/mL IL-1β for physioxia responsive donors. Data represent the fold change in the expression of pellets cultured under (**a**) 0 ng/mL and (**b**) 0.1 ng/mL IL-1β physioxia relative to the corresponding hyperoxic condition. Data represent mean ± S.D. of n = 4 physioxia responsive donors; * *p* < 0.05.

**Table 1 cells-08-00936-t001:** List of chondrogenic genes used in custom PCR plates.

Transcription Factors	Cartilage Matrix
L-SOX5 L-SOX6 SOX9	Aggrecan (ACAN) Chondromodulin-1 (LECT1) Cartilage oligomeric matrix protein (COMP) Collagen type I α1 (COL1A1) Collagen type II α1 (COL2A1) Collagen type VI α1 (COL6A1) Collagen type IX α1 (COL9A1) Collagen type XI α2 (COL11A2) Lysl Oxidase (LOX) Lubricin (PRG4) Matrillin-3 (MATN3) Perlecan (HSPG2) Versican (VCAN)
Transforming growth factor-β receptors	Hypertrophy
TGF-β receptor I (TGFBR1) TGF-β receptor II (TGFBR2)	Collagen type X α1 (COL10A1) MMP-9 MMP-13 ADAMTS-4 ADAMTS-5 RUNX2

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
