# Peer review of "Physioxia Has a Beneficial Effect on Cartilage Matrix Production in Interleukin-1 Beta-Inhibited Mesenchymal Stem Cell Chondrogenesis"

_cells, 2019, doi:10.3390/cells8080936_

Round 1
Reviewer 1 Report
Major point
1. The reason why the authors focused on the physioxia effect in ‘the presence of IL-1b’. The purpose of experiment was elusive. What means the mixture condition of physioxia and IL-1b?
2. In figure 1c, the donor’s MSCs mostly showed upregulated collagen II content in 0 IL-1b. It is difficult to agree with the statement “but no difference in collagen II content” at line 190 in page 8.
3. The results for selecting the IL-1b concentration would be better to place in the first part of result.
Minor point
1. Please adjust the letter size in figure 2f and figure 3f in the same size.
2. The statement at 331 line, Page 19 needs reference.
Author Response
Dear Scientific editor and reviewers,
We thank the editor and reviewers for their comments regarding the manuscript titled: “Physioxia has a beneficial effect on cartilage matrix production in interleukin-1 beta inhibited Mesenchymal Stem Cell chondrogenesis”. The reviewer comments have helped to further improve the manuscript and the authors hope that this new and revised version will be approved for publication in Cells special issue: Pathways to Cartilage and Bone destruction in arthritis.
In accordance with the guidelines of the journal, corrections to the manuscript have been made in bold and red. Additionally, this point-by-point letter to the reviewers answers all the corrections suggested by the reviewers and clarifies the amendments to the manuscript, including specific points where changes have been made in the text.
We hope that the revised review can proceed to publication in Cells.
Thanks for your time and consideration.
Yours sincerely,
Dr. Girish Pattappa

Reviewer 2 Report
Patappa and colleagues isolated MSCs from iliac crest bone marrow aspirates and differentiated the expanded cells under normoxic and hyperoxic conditions, in the presence or absence of IL-1b. The experiments are well presented but overall the paper lacks novelty: the principle findings are presented elsewhere, in some cases up to 10 years earlier.
The paper is well written and clear throughout, here I raise a few specific points:
1) I am a little unclear why patients undergoing knee surgery would have aspirates collected from iliac crest. If knee reconstruction required soft tissues from the hip could the authors please clarify this, and/or rectify the methods section.
2) Line 92, reference 39 does not describe the protocol, instead another paper giving a reference to where it was described. Please remove this reference chain and provide a specific reference or write the method out in full.
3) Line 98 - the CO2 and O2 atmospheric components are described but not the remaining components.
4) Line 109 - 50mg/ml ascorbic acid is almost 1000x higher than often used. Please check concentrations.
5) Line 177 - several typos in gene names in table 1
6) Please increase the text size of the scale bar in Figs 2b,d,f,3b,d,f,5a,b, 6c,d.
7) Fig 2e - please change error bars to be consistent with 2a,c.
8) In several situations the * value is define as p>0.05. Line 237, line 251. This is misleading considering that * value is defined as p<0.05 in the figure legend for fig 3. Please make this clear.
9) Please include a more precise description of the p-values in all cases. The Tukey post-hoc test can give the p-value that can be included.
10) The findings that Col10a1 expression is approx. 10-20-times higher in physioxia than normoxia (Fig 6a,b) but collagen X protein is visibly lower (Fig 6c) should be explored further. The explanation from line 282 is not sufficient. There are several plausible explanations, for example there may be a technical problem with fixation of the sections used for staining - since the physioxia sample was much larger, perhaps longer fixation time was necessary, and the antigen not preserved. Alternatively there could be a biological reason, that the hyperoxic cells had already reached hypertrophy and downregulated Col10a1 expression, whereas at the same time-point the cells in the physioxia pellet were actively undergoing hypertrophy, and therefore have elevated Col10a1 expression. Currently this discrepancy makes it difficult to interpret the data. Please establish the reason for this discrepancy, perhaps by analysing the pellets over a range of time-points.
11) Typos: 'physioxia' in line 278, "with", line 355.
12) Please reduce the discussion substantially. Most of the discussion is not directly relevant to the presented data, and dilutes the important points.
Author Response

(The authors gave the same response as above.)

Reviewer 3 Report
Pattappa et al concluded that physioxia has a beneficial effect on cartilage matrix production in interleukin-1 beta inhibited MSC chondrogenesis. Although sophisticated data were provided, MSC chondrogenesis is merely through IL-beta single pathway. Some main concerns are in the following.
Abstract:
-Background part is too long. Donor dependent responses mean what? Conclusion is vague. The statement of significance is not clear. Suggest rewrite the abstract.
Methods:
-There was no any data to identify the expression of surface marker in MSC (eg., CD34, CD45, CD105, CD73, CD166, CD99 and how many percentage of positive cells, respectively). Flow cytometry analysis may be reported. Please provide.
-Concentration-dependent hypoxia was not given.
-How to determine the additional concentrations of IL-1 beta? Please clarify.
-Why chose antibody from mouse, rabbit rather than human?
-Quantitative protein analysis should be added to confirm, rather than gene expression alone.
-The data by using two-way ANOVA should be reported by F, squared, actual P value.
Results:
-Images of fluorescein staining were not correlated to the quantitative data.
-There was no animal model to verify the mechanisms.
Discussion:
-It is too long to focus. For example, Lines 346-356 is regarding additional environmental stimuli. In fact, there is no any bioreactor or external mechanical stimuli in this study. Therefore, should be concise thoroughly and only focus on the core of present study. Also, Suggest subtitle given, as results part. Line 374-381 is limitation, please remove limitation part to the paragraph before conclusion.
-Clinical relevance is suggested to explain.
Author’s team group has cited this manuscript (the same title) in this published paper, Int J Mol Sci. 2019 Feb; 20(3): 484.
Author Response

(The authors gave the same response as above.)

Round 2
Reviewer 1 Report
The manuscript was revised properly.
Author Response
Dear reviewer,
Thanks for your kind response to our revised manuscript and we appreciate the improvements that were suggested, as this has greatly enhanced the manuscript.
Thanks for your kind input.
Best regards from the authors
Reviewer 2 Report
Thanks for addressing the comments in a point-by-point manner. Many of the minor point have been addressed in full but the more important aspects remain unclear. Please see my new comments in bold below:
1. I am a little unclear why patients undergoing knee surgery would have aspirates collected from iliac crest. If knee reconstruction required soft tissues from the hip could the authors please clarify this, and/or rectify the methods section.
We understand the reviewers requirement for clarification on the bone marrow aspirates and their localisation. Our group uses a bone graft taken from the iliac crest to augment autologous chondrocyte implantation (ACI) for the treatment of large and deep osteochondral defects. Thus, bone marrow was isolated during this procedure and according to our approved ethical guidelines. To clarify this, the methods sections has been amended to the following,
Page 4, Line 88: “Human bone marrow aspirates were isolated from the iliac crests of nine male patients (mean age: 30 + 11 years) undergoing knee arthroplasty and that required autologous bone grafting, sourced from the iliac crest for the treatment of deep osteochondral defects, following informed consent of the patients and using procedures that were approved by the local ethics committee (University Hospital Regensburg; Ethic approval no.: Nr. 01/135) [45].”.
Additional reference
45) Zellner J, Grechenig S, Pfeifer CG, Krutsch W, Koch M, Welsch G, Scherl M, Seitz J, Zeman F, Nerlich M, Angele P. Clinical and Radiological Regeneration of Large and Deep Osteochondral Defects of the Knee by Bone Augmentation Combined With Matrix-Guided Autologous Chondrocyte Transplantation. Am J Sports Med. 2017 Nov;45(13):3069-3080
Thanks
2. Line 92, reference 39 does not describe the protocol, instead another paper giving a reference to where it was described. Please remove this reference chain and provide a specific reference or write the method out in full.
We thank the reviewer for this important point within our methods section. The reference has been removed and been replaced with a full protocol for the isolation of the MSCs. The sentence(s) have been changed to the following,
Page 4, Line 97:”The mononuclear cell population from bone marrow aspirates was counted and then seeded into flasks at a density of 1 x 105 cells/cm2 in low glucose Dulbecco’s Modified Eagle medium (DMEM; Invitrogen, Karlsruhe, Germany) supplemented with 10% (v/v) foetal bovine serum (FBS; PAN Biotech, Aidenbach, Germany), 1% (v/v) penicillin-streptomycin (Invitrogen) and 5ng/ml basic fibroblastic growth factor (bFGF; Peprotech, Hamburg, Germany)”.
Thanks
3. Line 98 - the CO2 and O2 atmospheric components are described but not the remaining components.
The remaining component in the incubator is nitrogen. In the instance of culture under hyperoxia, this becomes 20% O2, 5% CO2 and 70% N2, whilst for physioxic culture, this changes to 2%O2, 5% CO2 and 93% N2. This has now been included in the Materials and Methods section to the following,
Page 4 Line 102: “Flasks were cultured in parallel in either a standard cell culture incubator at 20% oxygen, 5% CO2 and 70% N2 or a low oxygen incubator (ThermoFisher Scientific, Regensburg Germany) set at 2% oxygen, 5% CO2 and 93% N2.”
Thanks
4. Line 109 - 50mg/ml ascorbic acid is almost 1000x higher than often used. Please check concentrations.
The reviewer is correct in their point regarding the concentration of ascorbic acid. Indeed, this was a mistake in the original manuscript and has been corrected to 50μg/ml ascorbic acid.
OK
5. Line 177 - several typos in gene names in table 1.
The typos in the table 1 have been corrected in the table and indicated in red.
OK
6. Please increase the text size of the scale bar in Figs 2b,d,f,3b,d,f,5a,b, 6c,d.
The text size in the figures detailed have been increased to allow easier to visualise for the readers.
Thanks
7. Fig 2e - please change error bars to be consistent with 2a,c.
Error bars have been changed in figure 2e to ensure consistency with the rest of the figure.
Thanks
8. In several situations the * value is define as p>0.05. Line 237, line 251. This is misleading considering that * value is defined as p<0.05 in the figure legend for fig 3. Please make this clear.
We thank the reviewer for their comment regarding the p significance. To prevent misleading the reader, the * has been removed from the line 237 and 251, whilst no significant difference is defined as p > 0.05.
Thanks
9. Please include a more precise description of the p-values in all cases. The Tukey post-hoc test can give the p-value that can be included.
A precise p-value from the statistical tests and from the Tukey post hoc test has been included where appropriate within the main text of the manuscript.
Examples include,
Page 9, Line 220: “Examination of all donors indicated a significant difference in matrix production under physioxia, with respect to wet weight (p = 0.0004), GAG (p < 0.0001) and collagen II (p = 0.0078) content (Figure 2a-c; *p < 0.05)”
Page 9, Line 238: “In contrast, responders showed a significant increase in pellet wet weight and GAG (8-fold increase; both p < 0.0001) and collagen II content (3-fold increase; p = 0.0121) with respect to hyperoxia (Figure 4a, c, e; *p < 0.05)”
Thanks.
10. The findings that Col10a1 expression is approx. 10-20-times higher in physioxia than normoxia (Fig 6a,b) but collagen X protein is visibly lower (Fig 6c) should be explored further. The explanation from line 282 is not sufficient. There are several plausible explanations, for example there may be a technical problem with fixation of the sections used for staining - since the physioxia sample was much larger, perhaps longer fixation time was necessary, and the antigen not preserved. Alternatively there could be a biological reason, that the hyperoxic cells had already reached hypertrophy and downregulated Col10a1 expression, whereas at the same time-point the cells in the physioxia pellet were actively undergoing hypertrophy, and therefore have elevated Col10a1 expression. Currently this discrepancy makes it difficult to interpret the data.
Please establish the reason for this discrepancy, perhaps by analysing the pellets over a range of time-points.
The reviewer makes an excellent point and we thank them for their kind suggestions regarding the differences in COL10A1 mRNA expression and collagen X staining within the pellet. In response, we performed a PCR and collagen X immunostaining on day 14 pellets and compared them to the day 21 pellets presented in figure 6a and b. This was to observe whether there was a time course difference in expression.
In this instance, there was a reduction in COL10A1 expression in both control and IL-1β treated pellets under physioxia on day 14 compared with day 21, whilst there a significant downregulation in COL10A1 in IL-1β treated pellets on day 14 relative to hyperoxia. Immunohistochemical staining was similar to day 21, whereby there was more collagen X staining in the matrix in hyperoxia pellets compared to physioxia with or without IL-1β presence. Based on the present data and a previous publication, we feel that there is a delayed hypertrophic protein expression occurring in the physioxic MSC chondrogenic pellets. Supplementary figure 1 and the following paragraph have been added to the discussion to comment on this phenomena,
Page 19, Line 361: “We also examined pellets at an earlier time point on day 14 to understand whether there was a change in gene expression or immunostaining at an earlier time point (Supplementary figure 1). The analysis demonstrated that there was a reduced gene expression in COL10A1 on day 14 with or without IL-1β under physioxia compared to day 21 pellets. Histological staining demonstrated that there was minimal staining for physioxia cultured pellets independent of IL-1β presence, whereas there was collagen X staining in hyperoxic pellets, although lower compared to day 21. A previous study showed that protein expression of hypertrophy markers was observed only on day 28 under physioxia - indicating a delay in its expression, although reduced compared to hyperoxic chondrogenic pellets [50]. Thus, for the chondrogenic donors evaluated in this study, it can be postulated that there was a delay in the protein expression for hypertrophic markers and that later time points could lead to greater matrix deposition. Interestingly, in vivo studies show that physioxia preconditioned MSCs produce stable articular cartilage whereas hyperoxic MSCs became vascularized and formed bone upon implantation in a nude mouse model [41]”.
Additional reference
50) Gawlitta D, van Rijen MH, Schrijver EJ, Alblas J, Dhert WJ. Hypoxia impedes hypertrophic chondrogenesis of human multipotent stromal cells. Tissue Eng Part A. 2012 Oct;18(19-20):1957-66. j
Supplementary figure 1, The collagen X mRNA expression in (a) control (0ng/ml IL-1β) and (b) IL-1β treated pellets in physioxia responsive donors on day 14 and 21. Data represents fold change in expression of pellets cultured under physioxia relative to corresponding conditions under hyperoxia. Data represent mean + S.D. of n =4 physioxia responsive donors; *p < 0.05. Representative images of collagen X staining on (c) day 14 and (d) 21.
Thanks for adding an additional time-point but unfortunately the data still does not address the problem - why does the expression of chondrocyte hypertrophy markers in response to physioxia increase when analysed by RT-PCR but decrease when analysed by immunofluorescence? I realise that you have conflicting data, but these inexplicable findings are crucial to the entire manuscript, and the heading in the results section remains "Physioxia ... suppresses markers for hypertrophy" (line 288) despite these conflicting data. This needs to be addressed before publication. Some suggestions to confirm these protocols are: validation of PCR primers using mRNA isolated from human hypertrophic cells to use as a positive control; a new analysis using an entirely different pair of primers; validation of the immunostaining protocol using sections of human tissue to show that the antibody specifically labels hypertrophic zone - as type-X collagen is highly conserved, immunostaining of hypertrophic cells in other species, such as rodents, would also satisfactorily validate the staining protocol. The newly added discussion section relating to this point does not make this issue any clearer at all and should be removed.
11. Typos: 'physioxia' in line 278, "with", line 355.
We thank the reviewer for pointing out these typographical errors. The typos have been corrected and also throughout the manuscript.
Thanks
12. Please reduce the discussion substantially. Most of the discussion is not directly relevant to the presented data, and dilutes the important points
We understand the reviewer’s opinion on the discussion section of the manuscript. In response, the discussion has been substantially reduced and only the important points from the results have been evaluated in detail. Please see the new discussion with comments in red and bold for the amendments.
This section needs further improvement. Please include only the most crucial and relevant points, and edit the text entirely to produce a more concise discussion.
Author Response
1. Thanks for adding an additional time-point but unfortunately the data still does not address the problem - why does the expression of chondrocyte hypertrophy markers in response to physioxia increase when analysed by RT-PCR but decrease when analysed by immunofluorescence? I realise that you have conflicting data, but these inexplicable findings are crucial to the entire manuscript, and the heading in the results section remains "Physioxia ... suppresses markers for hypertrophy" (line 288) despite these conflicting data. This needs to be addressed before publication. Some suggestions to confirm these protocols are: validation of PCR primers using mRNA isolated from human hypertrophic cells to use as a positive control; a new analysis using an entirely different pair of primers; validation of the immunostaining protocol using sections of human tissue to show that the antibody specifically labels hypertrophic zone - as type-X collagen is highly conserved, immunostaining of hypertrophic cells in other species, such as rodents, would also satisfactorily validate the staining protocol. The newly added discussion section relating to this point does not make this issue any clearer at all and should be removed.
We thank the reviewer for the critical point regarding the hypertrophy data and we appreciate the suggestions for validating the expression of these markers. In this instance, we used the same antibody and protocol on rabbit articular cartilage and found positive staining for collagen X, specifically in the calcified cartilage layer. This has been added as a positive control to supplementary figure 1.
The collagen X PCR primer has been validated by our lab in a previous publication, whereby hypertrophy was induced in MSC chondrogenesis and demonstrated an upregulation in collagen X (reference - Karl et al, 2014; Tissue Eng. Part A Jan;20(1-2):178-88).
For the results section in the publication, the sub-section title has been changed to “Physioxia enhances gene expression of chondrogenesis associated markers and suppresses markers for late stage hypertrophy” (page 13; line 17) to emphasise that late stage hypertrophic markers, particularly RUNX2, are downregulated under physioxia. Collagen X and MMP13 are considered early stage markers and text has been amended accordingly.
In the discussion section, we prefer to keep this section, as it is an important finding to discuss. The paragraph has been modified to focus only on the current experimental results for the collagen X and MMP13 data. Please see page 16, lines 22-32
2. This section needs further improvement. Please include only the most crucial and relevant points, and edit the text entirely to produce a more concise discussion.
We thank the reviewer for the suggestion with respect to the discussion. The discussion has been reduced by 40% with only critical and relevant points described. Please see pages 16 – 19.

Reviewer 3 Report
Thank you for the revision.
Author Response

(The authors gave the same response as above.)
